# SELF-SUPERVISED LEARNING FACILITATES NEURAL REPRESENTATION STRUCTURES THAT CAN BE UNSUPERVISEDLY ALIGNED TO HUMAN BEHAVIORS

**Soh Takahashi**\*, **Masaru Sasaki**\*, **Ken Takeda & Masafumi Oizumi**
Graduate School of Arts and Science
University of Tokyo
Tokyo, Japan
`{soht121,m-sasaki,tkkentakeda1248,c-oizumi}@g.ecc.u-tokyo.ac.jp`

## ABSTRACT

The structure of perceived similarity between objects is crucial for understanding human object recognition. The acquisition of such a similarity structure during development is a pivotal question in cognitive science and neuroscience. While previous studies have focused on supervised learning guided by external teacher signals of object categories, the absence of such signals in early development prompts an exploration of the role of self-supervised learning. Self-supervised learning is thought to be the dominant mechanism for pre-linguistic learning, and supervised learning takes place after language is learned. Here, we compare the similarity structure of human object representations with the internal representations of a deep neural network model that underwent training through self-supervised contrastive learning, followed by supervised learning. To compare two similarity structures at the fine-item-level, we employed an unsupervised alignment approach using Gromov-Wasserstein Optimal Transport rather than the conventional supervised alignment approach known as Representational Similarity Analysis. We found that the model trained via self-supervised contrastive learning followed by supervised learning was more aligned with human behavior compared to models solely trained by supervised learning or self-supervised contrastive learning. We also found that at the level of coarse categories, the internal representation structure acquired through self-supervised learning alone was somewhat alignable to human behavior. These results suggest that self-supervised learning and its combination with supervised learning are effective in acquiring similarity structure that is unsupervisedly alignable to human behavior, offering potential mechanisms for the development of human object representations.

## 1 INTRODUCTION

Perceived similarity relationships between objects have been used to understand human object recognition, given the fundamental role of similarity in human cognition. This similarity relationship (or structure) can help explain how we form categories (Rosch & Mervis, 1975; Nosofsky, 1986; Goldstone, 1994) and how we generalize knowledge of known objects to novel objects (Shepard, 1987; Edelman, 1998). Moreover, similarity structure can be used to relate behavior, neural activity, and computational models, which is useful for understanding the neural substrate of human cognition (Kriegeskorte & Kievit, 2013). Recently, there have been several attempts to reveal the high-dimensional similarity structure of human object recognition in large-scale experiments involving a large number of subjects with a large number of objects (Roads & Love, 2021; Hebart et al., 2019). As a particular interest of this study, Hebart et al. (2023) presented the THINGS-dataset, a landmark dataset containing 4.7 million human similarity judgements on 1854 natural object images. They identified 66 interpretable dimensions that represent the structure of human similarity judgements (Fig.1a).

---

\*These authors contributed equally to this work

This large-scale analysis raises interest in how humans acquire such a similarity structure of objects during development. Fueled by successes in the artificial intelligence communities, the field of computational neuroscience has made progress on this question with goal-driven deep neural networks (DNNs) (Yamins & DiCarlo, 2016). Since the early days, supervised learning DNNs have been extensively studied and have achieved some success in modeling the human brain and behavior (Yamins et al., 2013; Kriegeskorte & Kievit, 2013; Yamins et al., 2014; Cichy et al., 2016; King et al., 2019; Muttenthaler & Hebart, 2021). Supervised learning is a framework in which explicit teacher signals of object categories guide the learning process. The internal representation structure acquired through supervised learning is expected to be organized according to the categories provided by the teacher signals, as schematically shown in Fig.1b below.

However, considering that explicit teacher signals are not always available during human development, especially in early infancy before language acquisition, there is a growing interest in exploring alternative mechanisms for acquiring the similarity structure of objects. Recent studies using self-supervised learning DNNs have made significant progress in addressing this challenge (Zhuang et al., 2021; Konkle & Alvarez, 2022; Geirhos et al., 2021). In contrast to supervised learning, self-supervised learning is a framework that does not require any explicit teacher signals. In this study, we specifically consider contrastive learning, which is one of the most popular self-supervised learning methods and has been studied as a potential mechanism of representation learning in humans and animals (Zhuang et al., 2021; Konkle & Alvarez, 2022; Millet et al., 2022; Nayebi et al., 2023). In self-supervised contrastive learning, a model is trained by attracting positive samples (augmented views of the same data) and repelling negative samples (augmented views of different data), as shown in the schematic of the internal representation structure in Fig. 1b above. As shown schematically in Fig.1b, the structure acquired through self-supervised learning is expected to be qualitatively different from that acquired through supervised learning.

Although most previous studies have focused solely on either self-supervised or supervised learning, the current study examines the combination of both and how it contributes to the formation of similarity structures. Considering that explicit teacher signals for humans are conveyed through language, self-supervised learning is thought to be the dominant mechanism for pre-linguistic learning, and supervised learning takes place after language is learned. To model this, we used a DNN model that underwent self-supervised contrastive learning followed by supervised learning, which is considered as finetuning, and compared its internal representations with the similarity structure of human object representations (Fig.1c).

To compare different similarity structures obtained from human behavior and DNN, Representational Similarity Analysis (RSA) has been widely used in cognitive science and neuroscience (Kriegeskorte et al., 2008). RSA is one of the supervised alignment methods (Kunz et al., 2021) which assumes that an element in one similarity structure (for example, the object 'dog') corresponds to the same element in another similarity structure (Fig.1d above). It then quantifies the degree of similarity between the different structures as the correlation between them. Although the correlation assessed by RSA can provide an overall similarity between two similarity structures, it has limitations in assessing more nuanced structural differences (Sasaki et al., 2023).

To address the limitations of RSA, we employed an unsupervised alignment approach using Gromov-Wasserstein Optimal Transport (GWOT) (Mémoli, 2011) (Fig.1d below). Unsupervised alignment provides a mapping between two similarity structures using only the internal distances (i.e. similarities) within each structure. This approach allows to assess the degree of structural correspondence between two similarity structures at the level of individual elements, which is not possible with conventional supervised alignment approach. Furthermore, based on the distributional hypothesis (Harris, 1954), where the meaning of a concept is determined by its relationship to other concepts, it makes sense to compare the similarity structure itself without using external labels.

Here, we aim to compare the internal representation structure acquired through self-supervised learning followed by supervised learning with the similarity structure of human object representations. For this purpose, we used the aforementioned large-scale dataset (THINGS dataset) of human similarity judgements of 1854 natural objects (Hebart et al. (2023); see Methods section for details). First, using GWOT unsupervised alignment, we compared the similarity structure of embedded image representations in a model trained only by self-supervised learning with the structure of human similarity judgements estimated from the THINGS-dataset. Next, we performed a similar unsupervised comparison between a model finetuned by further supervised learning and human similarity

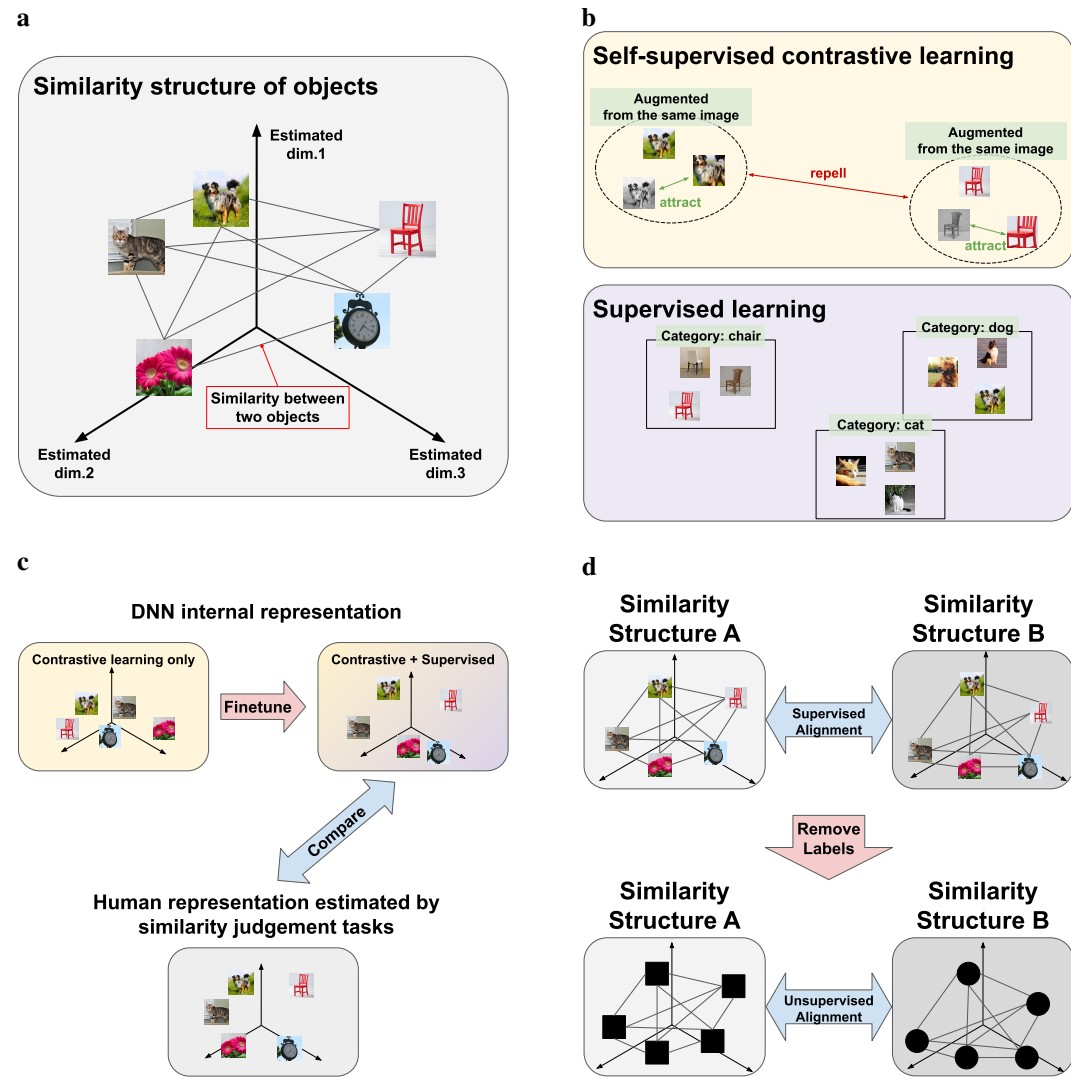

Figure 1:   Similarity structure framework and the overview of the study. (a) : Illustration of the similarity structure of human object representations. Each dimension is estimated using similarity judgement and the distance between each pair of objects represents estimated similarity. (b) : The schematic internal representation acquired by self-supervised contrastive learning (above) and supervised learning (below). (c) : Overview of the study. We compared the similarity structure of the DNN's internal representations with the similarity structure of human object representations. (d) : Illustration of supervised alignment (above) and unsupervised alignment (below).

judgements, which is our primary focus (Fig.1c). As a control, we also compare a model trained by supervised learning alone with human similarity judgements.

## 2   METHODS

### 2.1   DATASET OF HUMAN SIMILARITY JUDGEMENTS FOR NATURAL OBJECTS (THINGS)

As a representative of the similarity structures for natural objects in humans, we used THINGS dataset(Hebart et al., 2023) containing human similarity judgements for 1854 naturalistic objects. This dataset contains approximately 4.70 million human similarity judgements from about 12,000 participants collected through online crowdsourcing.

To assess the similarity between the 1854 natural images, they collected similarity judgements through behavioral experiments. In the experiments, participants performed an odd-one-out task, where they were presented with three naturalistic objects and asked to report which item in the triplet was the most dissimilar to the other two objects. The images used in the behavioral experiments were selected from the THINGS database (Hebart et al., 2019), which offers an extensive inventory of real-world objects encompassing both living and nonliving entities. For each concept, a single representative image was selected and used in the experiment.

To obtain the similarity structure among all 1854 natural images, we need to estimate psychological embeddings of each image, which is a quantitative characterization of objects as vectors in a multi-dimensional representational space. These embeddings were estimated to replicate human behavior in similarity assessments during behavioral experiments. In other words, the embeddings of two objects judged similar by humans were estimated to be close to each other, while the embeddings of two objects judged dissimilar were estimated to be far apart. We used the estimated 66-dimensional psychological embedding for 1854 objects provided by the dataset (https://osf.io/f5rn6/).

## 2.2 LEARNING OBJECTIVES

Here we explain the learning methods of deep neural networks that we investigated in this study.

### 2.2.1 SUPERVISED LEARNING

In supervised learning, a model is trained to predict the category of an object, which is provided as a teacher signal. The cross-entropy loss is used as the loss function:

$$L_{CE} = -\sum_{i=1}^{N} \sum_{j=1}^{C} y_{ij} \log \hat{y}_{ij} \tag{1}$$

where $N$ is the number of samples, $C$ is the number of classes, $y_{ij}$ is the ground truth label of the $i$-th sample for the $j$-th class, and $\hat{y}_{ij}$ is the predicted probability of the $i$-th sample for the $j$-th class.

### 2.2.2 SELF-SUPERVISED CONTRASTIVE LEARNING (SIMCLR)

SimCLR is a self-supervised learning method that uses contrastive loss to learn representations from unlabeled data (Chen et al. (2020a)). In SimCLR, a model is trained to maximize agreement between differently augmented views of the same data example via contrastive loss in latent space. The contrastive loss between a pair of positive examples $i, j$ (augmented from the same image) is given as follows:

$$L_{ij} = -\log \frac{\exp(\boldsymbol{z}_i^T \boldsymbol{z}_j / \tau)}{\sum_{k=1}^{2N} \mathbb{1}_{[k \neq i]} \exp(\boldsymbol{z}_i^T \boldsymbol{z}_k / \tau)} \tag{2}$$

where $N$ is the number of samples, $z_i$ and $z_j$ are augmented views of the same image, $\tau$ is the temperature parameter, and $\mathbb{1}_{[k \neq i]}$ is the indicator function that is 1 if $k \neq i$ and 0 otherwise.

### 2.2.3 FINETUNING

In this study, we refer to finetuning as a learning method in which self-supervised contrastive learning is followed by supervised learning. In finetuning, a model is first trained by self-supervised contrastive learning, and then further trained by supervised learning using the same dataset as the self-supervised contrastive learning.

## 2.3 MODELS

We used three models published in Chen et al. (2020b) (GitHub repository: https://github.com/google-research/simclr.git), all based on the ResNet101 2x architecture. Each model was trained using contrastive learning, finetuning, and supervised learning on the ImageNet dataset (Russakovsky et al. (2015)).The layer we used to define the Representational Dissimilarity Matrix (RDM) for each model is the avgpool layer after the last convolutional layer to average the features of them. TO preprocess the images to extract the embedding of the layer for each model, all the images were resized to the same resolution (224x224) and normalized over the RGB channels to the range 0-1.

## 2.4 Unsupervised alignment using Gromov-Wasserstein Optimal Transport

To compare two similarity structures in an unsupervised manner, i.e. without making any assumptions about the correspondence of objects between different similarity structures, we used the Gromov-Wasserstein Optimal Transport (GWOT) algorithm. The algorithm optimizes the Gromov-Wasserstein distance (GWD),

$$\text{GWD} = \min_{\Gamma} \sum_{i,j,k,l} (D_{ij} - D'_{kl})^2 \Gamma_{ik} \Gamma_{jl}, \tag{3}$$

which quantifies the correspondence between the two similarity structures. In our problem setting, $D_{ij}$ denotes the dissimilarity between object $i$ and $j$. Solving the GWD optimization problem yields an optimal transport matrix $\Gamma$, which represents the correspondence between the objects in the two similarity structures. An element of the matrix $\Gamma_{ik}$ can be interpreted as the "probability" that the $i$-th object in the first similarity structure corresponds to the $k$-th object in the second similarity structure.

Efficient optimization of GWD can be achieved by adding an entropy-regularization term, $H(\Gamma)$ to the objective function:

$$\text{GWD}(\Gamma) = \min_{\Gamma} \sum_{i,j,k,l} (D_{ij} - D'_{kl})^2 \Gamma_{ik} \Gamma_{jl} + \varepsilon H(\Gamma), \tag{4}$$

This addition has been proven to improve optimization efficiency (Peyré et al. (2016)). Since the optimization problem in Eq.4 is non-convex, we performed hyperparameter tuning on $\varepsilon$ and performed random initialization of the matrix $\Gamma$ to find good local minima.

To evaluate the degree of agreement between two similarity structures, we calculated the matching rate between the two similarity structures using object classes. For each object, we consider it as a match if the transportation plan assigns the highest probability between the same objects in the two similarity structures. We refer to this matching rate as the top1 matching rate. We also consider the category level matching rate, which we count as a match if the transportation plan assigns the highest probability between the same coarse categories in the two similarity structures.

## 3 Results

### 3.1 GWOT alignment with self-supervised contrastive learning model

First, we show the result of the GWOT alignment between the similarity structures of human behavior and the model trained by self-supervised contrastive learning only. The RDMs of human behavior and the model are shown in Fig.2a. As described in the Methods section, we performed over 500 optimization iterations with different values of the hyperparameter $\epsilon$ and selected the local minimum with the lowest GWD as the optimal solution (shown as the red circle in Fig.2b).

The optimal transport matrix corresponding to the optimal solution is shown in Fig.2c. This optimal transport matrix significantly deviates from being a diagonal matrix, meaning that human objects are associated with different objects in the contrastive learning model. To quantitatively evaluate the degree of the fine-item level correspondence, we computed a top1 matching rate of 0.16%, which is almost at chance level (0.05%).

In addition to the fine-item level matching, we also investigated the correspondences at the coarse-categorical level. Fig.2d shows an enlarged view of some of the coarse categories of the optimal transport matrix in Fig.2c. As can be seen in this enlarged view of the optimal transport matrix, objects in certain categories, such as food and tool, are transported in the same coarse category, even though they do not match at the fine-item level.

In sum, these results indicate that contrastive learning alone may acquire a similarity structure that is somewhat aligned with human behavior, but not at the fine-item level.

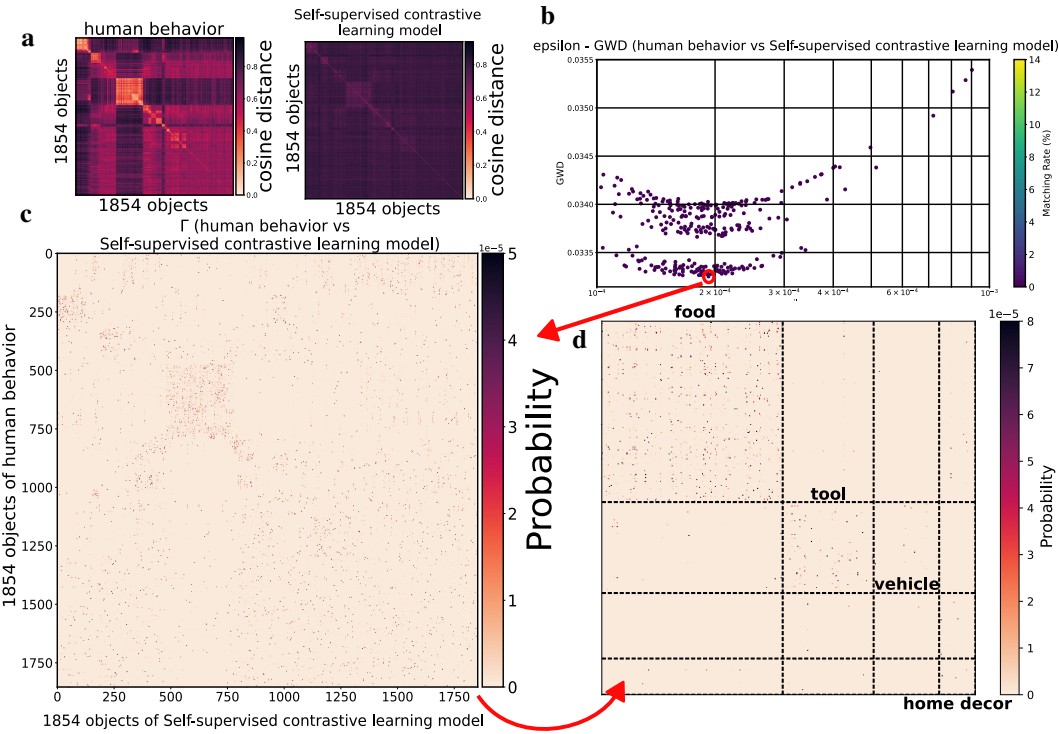

Figure 2: GWOT alignment between human and contrastive learning model. (a) : The RDM of human behavior and contrastive learning model for 1854 images of THINGS dataset. (b) : Relationship between GWD and hyperparameter $\varepsilon$. Color represents top1 matching rate. Each dot in this figure represents the result of the optimization of the OT matrix with the hyperparameter $\epsilon$. The OT matrix shown above is the result of the minimum GWD in this figure. (c) : The OT matrix between the RDM of human behavior and contrastive learning model for 1854 images of THINGS dataset. (d) : Enlarged view of the OT matrix in (c), specifically for some coarse categories (food, tool, vehicle, and home decor)

## 3.2 GWOT ALIGNMENT WITH FINETUNED MODEL

Next, we show the result of the GWOT alignment between the similarity structures of human behavior and the model trained by contrastive learning and then finetuned by supervised learning. The results are presented in the same manner as Fig.2.

In contrast to the case of the contrastive learning model, we can see that the optimal transport matrix is roughly diagonal (Fig.3c, d), i.e. the diagonal elements or elements in the neighborhood tend to have high values. This roughly diagonal appearance means that similar objects correspond to each other between human behavior and the finetuned model. The top1 matching rate was 13.97%, which is significantly higher than that of the contrastive learning model (0.16%) and chance level (0.05%). As can be seen in Fig.3b, the local minima with low GWD tend to have high matching rates, which is necessary for successful unsupervised alignment. These results indicate that supervised learning after contrastive learning is effective in acquiring a similarity structure that can be unsupervised aligned to human behavior.

## 3.3 GWOT ALIGNMENT WITH SUPERVISED MODEL

For comparison with the finetuned model, we show the result of the GWOT alignment between the similarity structures of human behavior and the model trained by supervised learning only. The results are presented in the same manner as Fig.2 and Fig.3.

The top1 matching rate was 7.77%, which is lower than that of the finetuned model (13.96%). This difference in the top1 matching rate can be seen as the difference in the values of the diagonal

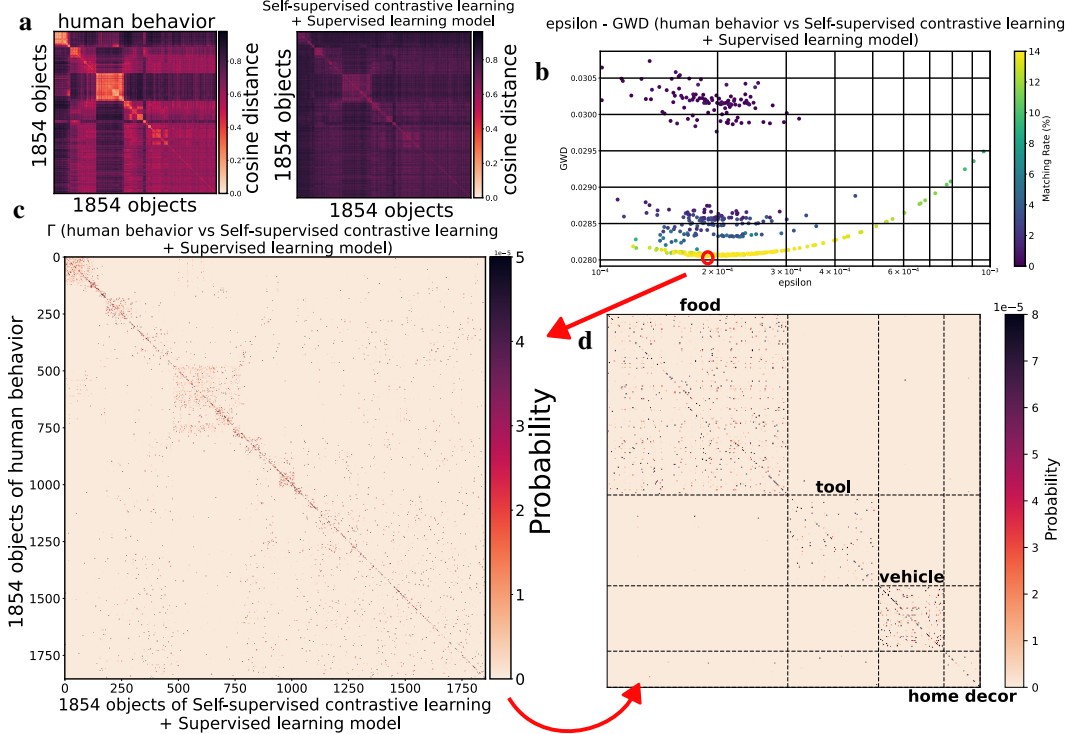

Figure 3: GWOT alignment between human and finetuned model. (a) : The RDM of human behavior and finetuned model for 1854 images of THINGS dataset. (b) : Relationship between GWD and hyperparameter $\varepsilon$. Color represents top1 matching rate. Each dot in this figure represents the result of the optimization of the OT matrix with the hyperparameter $\epsilon$. The OT matrix shown above is the result of the minimum GWD in this figure. (c) : The OT matrix between the RDM of human behavior and finetuned model for 1854 images of THINGS dataset. (d) : Enlarged view of the OT matrix in (c), specifically for some coarse categories (food, tool, vehicle, and home decor)

elements in the optimal transport matrices (compare Fig.3c, d and Fig.4c, d). Comparing Fig.3b and Fig.4b, we can see that the tendency for the matching rate to be higher when the GWD is smaller is weaker than in the case of the finetuned model. These results suggest the importance of prior self-supervised contrastive learning, rather than just supervised learning, in acquiring similarity structure that can be unsupervisedly aligned to human behavior.

### 3.4 COMPARISON BETWEEN DNNS AND HUMAN BEHAVIOR

Finally, we show the summarized results of GWOT top1 matching rate, category matching rate, and RSA value for each model to THINGS human behavior data in Fig.5. For all metrics, the finetuned model has the highest similarity to human behavior. We also note that while top1 matching rate between the contrastive learning model and human behavior is almost at chance level (Fig.5a), category matching rate is 16.34%, which is significantly higher than chance level (3.57%) (Fig.5b). This result implies that the contrastive learning alone can acquire the similarity structure, which is somewhat alignable to human behavior at the level of coarse categories.

Comparing the results of GWOT top1 matching rate (Fig.5a) and RSA value (Fig.5c), we can see that the differences between the contrastive learning model and the other models are more prominently captured in GWOT top1 matching rate than in RSA value. This result suggests that the GWOT unsupervised alignment method is able to capture more nuanced structural differences that are not captured as well by RSA.

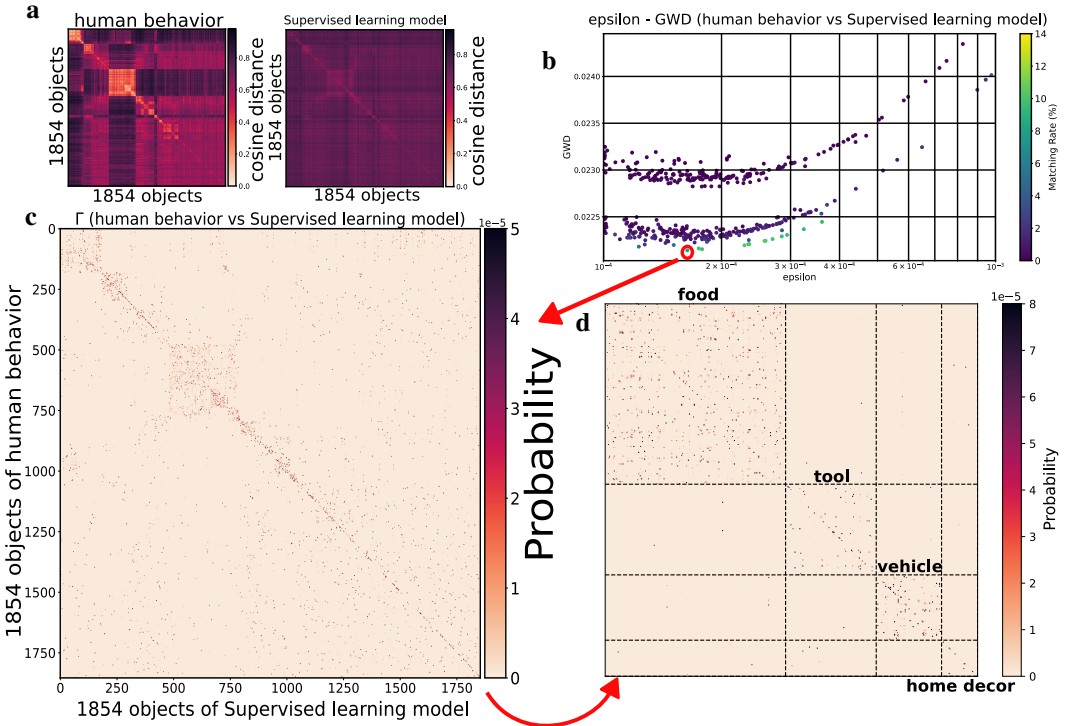

Figure 4: GWOT alignment between human and supervised model. (a) : The RDM of human behavior and supervised model for 1854 images of THINGS dataset. (b) : Relationship between GWD and hyperparameter $\varepsilon$. Color represents top1 matching rate. Each dot in this figure represents the result of the optimization of the OT matrix with the hyperparameter $\epsilon$. The OT matrix shown above is the result of the minimum GWD in this figure. (c) : The OT matrix between the RDM of human behavior and supervised model for 1854 images of THINGS dataset. (d) : Enlarged view of the OT matrix in (c), specifically for some coarse categories (food, tool, vehicle, and home decor)

## 4 DISCUSSION

In this study, we investigated the internal representation structure acquired through self-supervised contrastive learning followed by supervised learning (finetuning) and compared it with the similarity structure of human object representations using THINGS dataset. To compare two similarity structures at the level of individual objects, we employed an unsupervised alignment approach using Gromov-Wasserstein Optimal Transport. We found that the finetuned model was more aligned with human behavior compared to models solely trained by supervised learning or self-supervised contrastive learning. We also found that at the level of coarse categories, the internal representation structure acquired through self-supervised learning alone was somewhat aligned with human behavior. These results suggest that self-supervised learning and its combination with supervised learning provide potential mechanisms for the development of human object representations.

Furthermore, we demonstrated that the unsupervised alignment method based on GWOT can capture fine-item-level structural differences that cannot be captured by RSA. For instance, we found that the similarity structure of contrastive learning model was not alignable to human behavior at the level of individual object. However, it was alignable to human behavior at the coarse category level. This finding was made possible by the unsupervised alignment method. In addition, the difference between models in ease of alignment with human behavior is captured more prominently in GWOT unsupervised alignment than in RSA, suggesting that unsupervised alignment can capture more nuanced structural differences than RSA.

It is important to note that this study only tested a limited number of models. In order to generalize our findings, it is important to compare models and human behavior using a wide range of models. This includes different model architectures, training data sets, and learning methods. Specifically,

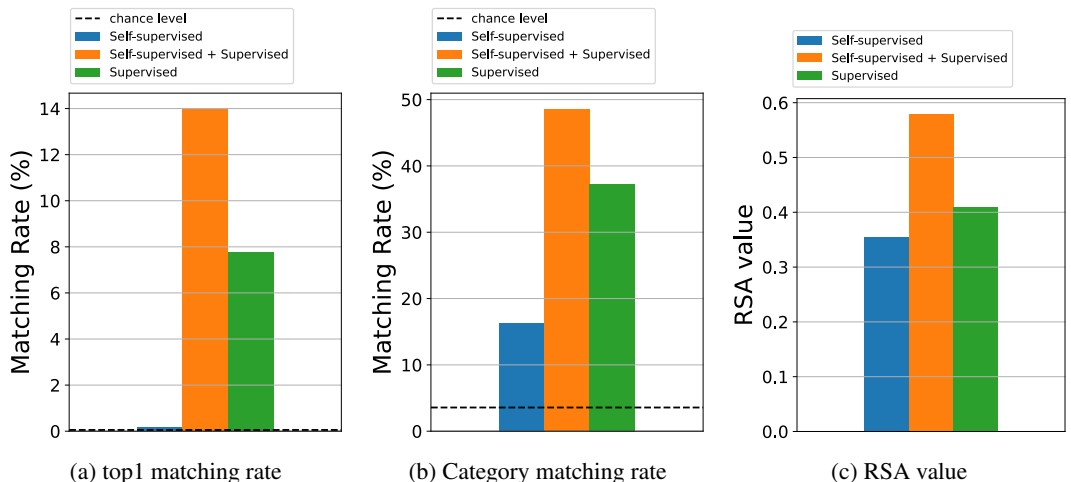

Figure 5: Comparison of the top1 matching rate, category matching rate and RSA value for each model to THINGS behavior data. (a) : Top1 matching rate for each model. (b) : Category matching rate for each model. (c) : RSA value for each model.

although this study focused on contrastive learning among self-supervised learning methods, it is important to use self-supervised learning methods that are more biologically plausible, such as contrastive predictive coding (Oord et al. (2018)), which is consistent with predictive coding theory in neuroscience (Rao & Ballard (1999); Friston (2005)).

While we focused on image input, it is also important to extend the self-supervised learning framework to multimodal input, including audio, image, and text. This multimodal approach is more consistent with the human learning environment. Multimodal machine learning methods, such as CLIP (Radford et al., 2021), have recently made significant progress. Some studies have already begun to investigate multimodal DNNs as models of human development (Vong et al., 2024). In future work, it is important to compare the similarity structure of multimodal DNNs trained via self-supervised and supervised learning with human behavior or brain responses.

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
