# OpenReview forum: "Self-supervised learning facilitates neural representation structures that can be unsupervisedly aligned to human behaviors"
_ICLR.cc/2024/Workshop/Re-Align — ICLR 2024 Workshop Re-Align Poster_

### Official Review · Reviewer_hgbR · 2024-02-22
**Review of "Self-supervised learning facilitates neural representation structures that can be unsupervisedly aligned to human behaviors"**

**Rating:** 2
**Fit:** 3
**Confidence:** 2

**Workshop Review:**

## Summary
This paper investigates Gromov-Wasserstein Optimal Transport (GWOT) as an alignment technique for aligning ML models to human similarity judgement data from the THINGS dataset.Three settings are compared: supervised training, SSL training, and a combination (SSL+supervised fine-tuning). The authors report that the last setting leads to the highest alignment using GWOT (in relative terms). In absolute terms, alignment is fairly low / there is a fair bit of room for improving on the reported metrics. Overall, this is a nice workshop submission and I am in favor of acceptance.

## Strengths
- Clear motivation
- the paper is well written & explained important concepts in an accessible way
- the three main figures (Figs 2--4), along with their similar structure, help a lot in making the results accessible

## Weaknesses
- GWOT is a new choice of alignment method. I am somewhat skeptical about the method. I'll try to explain it with a metaphor: let's say you're sitting in an airplane; the airplane is full. Before takeoff the captain makes an announcement that all passenger need to switch to a different airplane. Everyone gets out and tries to get their original seat in the new airplane, but in the rush a lot of people don't remember exactly where they were sitting so they just sit down close to where they think they were. A few people also have really poor memory and more or less randomly sit somewhere. Now at that point even if you remember exactly which seat you were originally sitting on, you may find that your seat is taken already and you need to sit somewhere else. As a result, lots of people end up sitting in different places compared to before - even though everyone is trying to sit where they were sitting previously... As a result, I'm not sure whether GWOT is a great choice of alignment method. Before introducing a new method in this context and trusting its conclusions, I would have loved to see for instance some experiments on controlled synthetically generated data comparing RSA and GWOT to make sure we have a good intuition about GWOT and can trust its results. This seems like a bit opportunity for improvement in future iterations of this paper.
- as a consequence, I would recommend reflecting the uncertainty about GWOT reliability in the discussion and perhaps adjust some sentences like "the difference between models in ease of alignment with human behavior is captured more prominently in GWOT unsupervised alignment than in RSA, suggesting that unsupervised  alignment can capture more nuanced structural differences than RSA." to reflect the fact that we currently don't know whether that's due to GWOT intricacies or reflecting underlying patterns in the data that can't be captured as well with RSA.

## Questions & MISC
- I may have missed it, but what are the ImageNet validation accuracies for the investigated base models, and are they comparable to the linear read-out accuracy of the SimCLR model? Differences in accuracies between models might influence the results.
- "These results suggest that self-supervised learning and its combination with supervised learning are effective in acquiring similarity structure that is unsupervisedly alignable to human behavior, offering potential mechanisms for the development of human object representations." - Do you mean human-like object representations?
- the paper length of 8.5 pages seems a tad long currently; a few sections might benefit from shortening. I may be biased since my background is ML; for a ML paper / venue it's a bit uncommon to read the first five pages before getting to the core of the paper (the results). Writing boils down to personal preference though.
- as a control experiment, have you considered starting with supervised training and then switching to SSL training (i.e. the reverse of the setting that works best)?

## Minor
- page 2 mentions figure above, but the figure is on the next page (below)
- "limitations in assessing more nuanced structural differences": expand? The motivation for using GWOT over RSA seems central to the paper, so this part could be expanded
- Figure 1d: I found the difference between the top and the bottom part of the figure a bit confusing, as it mentions 'remove labels' but also removes the images themselves, which is counter-intuitive (replacing them with black shapes)
- In section 2, rather than stating the supervised and SimCLR training objectives (which are well-known), it might help to add a bit more detail on fine-tuning: How many epochs? Learning rate? Etc.
- page 4 typo TO preprocess
- nit: in LaTeX, quotes are started via the `` symbol not the '' symbol
- p. 9 typos in citation formatting, leading to double brackets

**Reason For Not Giving Higher Score:**

see weaknesses discussed above

**Reason For Not Giving Lower Score:**

see strengths discussed above

**Reviewer Domain:**

machine learning

---

### Official Review · Reviewer_HQpH · 2024-02-23
**Joint self-supervised and supervised training yields representations aligned with human behavior**

**Rating:** 3
**Fit:** 3
**Confidence:** 2

**Workshop Review:**

The authors investigate how models trained using self-supervised, supervised, or both objectives align with human judgments of odd-one-out objects. The authors make use of an unsupervised alignment approach (Gromov-Wasserstein Optimal Transport). The paper has a strong premise and is creative, yet simple. The motivation is solid and has a clear topical fit to the workshop. Most of my questions were nicely answered throughout the paper.

The results show that the model trained on first unsupervised learning, thereafter supervised learning, best matches human judgments (using the GWOT metric). The results are also shown using a more standard RSA metric which is a nice complement to the GWOT results. It would be interesting to better understand where objects are mismatched between the two systems of interest (model and human), but could clearly be a paper in itself.

A few questions:
- What happens in the GWOT alignment if the same object in matrix A is matched several objects in matrix B, i.e., the one-to-many scenario?
- Is there a meaningful human-to-human "ceiling" / GWOT alignment value to be computed based on the behavioral dataset?
- The statements about "self-supervised learning is thought to be the dominant mechanism for pre-linguistic learning, and supervised learning takes place after language is learned" could use some references.

**Reason For Not Giving Higher Score:**

N/A

**Reason For Not Giving Lower Score:**

- Clear writing, good premise and execution.

**Reviewer Domain:**

neuroscience

---

### Official Review · Reviewer_Rt34 · 2024-02-25
**Empiric findings that SSL improves on supervised learning for unsupervised alignment with GWOT of neural networks to human data -- interesting but superficial**

**Rating:** 2
**Fit:** 3
**Confidence:** 3

**Workshop Review:**

Clarity
* Paper is generally clearly written and easy to read. There is some repetitiveness and use of vague language, with lack of important details -- I feel this paper could easily be half the length.
* I think it would benefit from more equations and explanation of techniques, as these are not always clear -- e.g. equation for matching rate in section 2.4 (this is usually given, as seen in other related papers), and full explanation of training (as noted in correctness below)
* There are some typos and grammatical errors, as well as a lot of repetitiveness that detracts from the overall readability. For example: 2.1 para 1 spacing (references run into text); 2.3 capitalisation errors; 2.4 para 1 incorrect use of quotation marks; figures with different label sizes and missing x-labels (e.g. F2(b) and others); 3.2+3.3 "unsupervised(ly) aligned" should be rephrased.
* Fig 5 could benefit from a logarithmic y-scale.

Correctness
* Lacking important details. GWOT not well explained.
* Comparing to a similar paper that uses GWOT for unsup alignment in neuroscience with several case examples (Sasaki et al., 2023 -- it is cited in this paper), there are many details (arguably similar) missing in this paper, in particular for the optimisation procedure and hyperparam tuning.
* Missing key equations e.g. 2.4 -- matching rate.
* We need more details about the hyperparam tuning used.
* Are "500 optimisation iterations" (3.1) enough -- any guarantees on this? Should we expect this to be the same across techniques? Arguably not surprising that the SSL+sup technique does better, but this is not discussed. It seems interesting that this is closer to supervised than SSL alone, though, which is also not discussed.
* 3.2 "roughly diagonal" -- should be able to quantify this using another comparison metric between approaches.

Novelty
* Overall seems empirical results of using SSL+sup for a DNN representation and then comparing to human data are strictly unique, but the methods are not. In particular, the novelty of using GWOT with unsupervised alignment for comparing representations in biological systems is oversold.
* One related paper on GWOT for unsupervised alignment in neuroscience by Sasaki et al., 2023 is mentioned but does not get much attention. The methods there seem to be similar (for the GWOT alignment part that is -- again, it seems some details are missing, but the RDMs from each paper look similar (figure 4 in the Sasaki paper)). The authors could consider comparing the human dataset results to Sasaki et al., which compares these in two (human) subgroups, and at least explore the comments on RSA made in this paper further, including in the discussion (para 2 -- others have made this observation, but are not cited).

Interest
* I think this work is of interest to the community, and there are limited works comparing DNNs to neural findings. However, besides the empiric result, this work is relatively superficial and provides minimal mechanistic understanding. It is particularly interesting that supervised alone outperforms SSL alone, especially in light of the introduction, but this is not explored at all.
* Error bars from multiple runs would be nice (for direct comparison of top1 matching rates, i.e. is 7% really that different from 13% or just a spurious result up to iterations/noise? -- perhaps the landscape just finds more local minima (and if so, why?)).
* Similarly, is there any way to make sup perform as well as SSL+sup? For example, more iterations, broader optimisation sweep... Perhaps the operational range of epsilon is vastly different? These limitations are not discussed at all, and seem interesting.

**Reason For Not Giving Higher Score:**

Regarding fit: Please see comments above. I think this paper is borderline accept level, because of some key missing details and incomplete discussion, although the result itself is novel and interesting. Therefore a higher score would not be appropriate. In deliberating between 1 and 2, I think the clear topical fit and interesting empiric result means that it is worth showcasing at the workshop.

Regarding fit: N/A

**Reason For Not Giving Lower Score:**

Regarding fit: N/A

Clearly fits "Can representational alignment tell us if AI systems use the same strategies to solve tasks as humans do?", if lacking on some details that would aid further work in this domain (see review above)

**Reviewer Domain:**

neuroscience

---

### Decision · Program_Chairs · 2024-03-02

Accept (Poster)